# Evaluation of Food Waste Prevention Measures— The Use of Fish Products in the Food Service Sector

**Yanne Goossens *****, Thomas G. Schmidt and Manuela Kuntscher**

Thünen Institute of Rural Studies, Bundesallee 64, 38116 Braunschweig, Germany;
thomas.schmidt@thuenen.de (T.G.S.); manuela.kuntscher@thuenen.de (M.K.)

**\*** Correspondence: yanne.goossens@thuenen.de

**Abstract:** This study presents two food waste prevention measures focusing on the interface between the food service sector and its food suppliers. Through a case study on procuring salmon by a hotel kitchen, the use of food products with different convenience grades is examined. The convenience grade of the fish bought (whole salmon, fillets or portions) determines where along the food chain filleting and/or portioning takes place and thus where food waste from cut-offs occurs. To reduce food waste, we propose purchasing filleted or portioned salmon rather than whole salmon. For both measures, effectiveness is calculated by looking at food waste reductions along the food chain, achieved by a better use of filleting and portioning cut-offs. Next, sustainability across the environmental, economic and social dimension is evaluated by calculating (a) avoided embodied environmental impacts and economic costs, (b) avoided food waste disposal environmental impacts and economic costs and (c) environmental, economic and social impacts and costs associated with implementing the measures. Purchasing fillets or portions instead of whole salmon leads to food waste reductions of −89% and −94%, respectively. The interventions further lead to net climate change impact savings along the salmon chain of −16% (fillets) and −18% (portions). Whereas the kitchen saves costs when switching to fillets (−13%), a switch to portions generates additional net costs (+5%). On a social level, no effects could be determined based on the information available. However, good filleting skills would no longer be needed in the kitchen and a time consuming preparation can be sourced out.

**Keywords:** food waste; measure; sustainability evaluation; LCA; costs; fish processing; out-of-home; food service

## 1. Introduction

### 1.1. Setting the Scene

According to the FAO, about one third of all edible food mass intended for human consumption—or about 1.3 billion tonnes per year—is lost or wasted throughout the food chain. Including inedible parts, the global volume of food wastage goes up to 1.6 billion tonnes [1,2]. At the European level, food waste was estimated at 88 million tonnes for 2012, comprising both edible and inedible parts of food and representing about 20% of all the food produced in the EU [3,4]. It should be noted that food or parts of food removed from the supply chain and valorised (for example as animal feed) is not considered as food waste in the EU calculations and is thus not included in the EU food waste volumes, whereas it is part of the FAO data [1,3].

Food gets lost or wasted throughout the entire food chain. The last step along the chain, the consumption stage, includes both private households and the food service sector. The food service sector can be divided into (1) commercial food services such as restaurants and hotels and (2) food services within the healthcare, education, public or business sector, where the primary focus of serving

food is to provide a service to patients, staff or students [5–8]. In the EU, the food service sector was responsible for about 12% of the 88 million tonnes of food waste in 2012 [3].

Target 12.3 of the United Nations Sustainable Development Goals (SDGs), adopted in 2015, calls for halving per capita global food waste at the retail and consumer levels and for reducing food losses along the production and supply chains, including post-harvest losses, by 2030 [9,10]. To achieve this target, many measures have been proposed and implemented so far. In a literature review, Goossens et al. looked at methodologies to evaluate food waste prevention measures, concluding that economic, environmental or social assessments are often incomplete or missing [11]. This prevents practitioners and decision-makers from prioritising food waste measures. Building on recent developments at the EU level, such as the report from Caldeira et al. [12], Goossens et al. put forward a sustainability assessment framework to evaluate food waste prevention measures [11]. To illustrate its use and emphasise the additional value of the proposed framework, the present paper applies this framework to a case study in the food service sector—more particularly, to a food waste measure applied in hotel kitchens.

### 1.2. Use of Convenience Food Products in the Food Service Sector

Both individual consumers and the food service sector are increasingly using so-called convenience food to reduce the workload in the kitchen. The convenience or processing grade of fresh or frozen food products differs widely, as there is a wide range of products on the market. A categorisation into convenience grades is shown in Table 1.

**Table 1.** Categorisation of food products into convenience grades and example food products. Based on [13–15].

| Convenience Grade | Description | Example Food Products |
|:---:|:---:|:---:|
| 0 | No-convenience food | All original raw food |
| 1 | Kitchen-ready food, inedible parts have been removed | Boneless meat, filleted fish, cleaned vegetables |
| 2 | Ready-to-cook foods | Fish or meat portions, frozen vegetables or fruit, pre-baked bread |
| 3 | Prepared foods, often require heating and adding other ingredients | Powders for mashed potatoes, desserts, sauces or soups |
| 4 | Food ready for regeneration, only needs to be heated | Individual ready-made components (such as sauces) or ready-made menus (such as stews) |
| 5 | Ready-to-serve foods, can be consumed after packaging removal/opening | Cold sauces, smoothies and ready-to-eat salads and desserts |

### 1.3. Food Waste and Sustainability Aspects Related to Using Convenience Food Products

The convenience grade of a food product is determined by where along the chain food processing or preparation takes place: at the level of the food manufacturer/supplier or at the level of the consumer. As such, the convenience grade of a product determines where food trimmings or cut-offs are generated. Since these food trimmings often end up as preparation waste in a large-scale kitchen, the convenience grade of a product affects the amount of preparation waste generated. In the literature, the share of preparation waste to the total amount of food waste in large-scale kitchens varies. According to Engström and Carlsson-Kanyama, storage and preparation waste contribute to only 20%, whereas Cerutti et al. and WRAP report 42% and 45%, respectively [16–18].

Several studies look at how the centralisation of food processing or preparation affects the environmental impacts associated with food consumption by households [19–26]. Only a limited

amount of these studies also clearly specify how the level of convenience affects food waste quantities and the chosen food waste management route [23,24,27]. According to these few studies, the switch to food with a higher convenience grade (such as ready-made meals or meal kits with pre-portioned ingredients) could be seen as an effective food waste measure for households, often leading to environmental savings as well.

When it comes to the food service sector, various studies can be found looking into the environmental impacts of food services [28], the impacts associated with food waste arising in this sector [29] and how a food service can reduce greenhouse gases and/or resource use [30,31]. Other studies focus on changing procurement practices in order to reduce impacts, such as improved planning and forecasting [32] or switching to buying seasonal or organic food [17,30] and reducing meat consumption [17]. To the authors' knowledge, only one study looked—in the sidelines—into the issue of buying convenience products in the food service sector and how this affects greenhouse gas emissions [30]. That particular study concluded that serving freshly prepared potato mash was associated with lower greenhouse gas emissions than mash prepared using potato powder. Furthermore, the authors propose using fresh ingredients rather than convenience products to reduce impacts. WRAP further suggests procuring a part of the menu in pre-prepared format to reduce preparation waste in the kitchen [33]. Similarly, Göbel et al. suggest using pre-trimmed produce, meats and fish, as well as purchasing dough pieces for bakery products to be baked on the spot in order to reduce food waste [34]. However, no data on the effectiveness or sustainability of such food waste measures in the food service sector could be found.

The literature further shows that the use of convenience products can save labour time and costs, despite eventual higher product prices. Additionally, its use lowers the needs for specific training of staff, as convenience foods may be easier to prepare and cook than cooking from scratch. Furthermore, the use of convenience products could ensure more consistency in the products used in the kitchen and subsequently served to guests [35–37]. On the other hand, the increased use of convenience foods in the food service sector may be associated with the use of other, and potentially more, packaging materials. Additionally, using these products may lead to a deskilling of staff, reduce opportunities for creativity and lower staff motivation [35,37,38]. However, no case studies could be found giving exact numbers on how the use of convenience foods in a commercial kitchen affects a food service business.

## 1.4. Goal and Focus of this Study

The present study addresses the literature gap on using food products of various convenience grades in the food service sector, its effects on food waste and the associated sustainability. This study focusses on the interface between the food service sector and its food suppliers, investigating food products with convenience grades 0, 1 and 2 (Table 1). More specifically, through a case study in a hotel kitchen, this paper examines two food waste measures focusing on filleting and portioning of Atlantic salmon and its stage in the food chain.

In earlier times, large-scale kitchens tended to buy whole fish (convenience grade 0) from its suppliers, after which the fish were filleted and portioned in the kitchen. Recently, due to a shortage of staff with good filleting skills, a trend can be observed towards buying filleted or even portioned fish (convenience grades 1 or 2 respectively). In this case, filleting—and possibly even portioning—no longer takes place in the kitchen, but at the manufacturing site of the fish supplier. When filleting and/or portioning takes place in a large-scale kitchen, most filleting and portioning cut-offs are thrown in the bin and end up as food waste. In case filleting and/or portioning takes place at the supplier manufacturing site, the cut-offs are centralised and used for internal or external processing. As such, rather than shifting food waste to another food supply chain stage, these fish by-products no longer become waste, as they are used for other human consumption purposes or valorised as animal feed, for example.

The food waste measures under study in this paper refer to a hotel kitchen no longer procuring salmon with a convenience grade 0 (whole salmon), but instead purchasing salmon with a convenience

grade of 1 (fillets) or 2 (portions). Following a better use of and an increased valorisation of fish by-products, the food waste measures are expected to contribute to reducing salmon food waste along the chain. Food waste savings are in turn expected to lead to environmental benefits. The purchase of filleted or portioned salmon, however, comes at a higher price per kilogram than whole salmon, leading to many kitchen chefs hesitating to make this switch. Yet if filleting and/or portioning are taken up by the kitchen staff, significant labour costs are associated with this highly specialised skill. For a food service business, it is thus not always clear which option is preferable. This paper therefore assesses the extent to which purchasing salmon with a higher convenience grade can reduce salmon food waste along the chain and improve sustainability across the environmental, economic and social dimension.

## 2. Materials and Methods

### 2.1. Case Study Description and Inventory Data

The present case study was set up following a collaboration with Maritim Hotelgesellschaft mbH, a major hotel group in Germany, and Deutsche See GmbH, one of the main fish suppliers in Germany, in the context of the research project ELoFoS (Efficient Lowering of Food Waste in the Out-of-Home Sector; https://elofos.de).

The case study in this paper focusses on the supplier-kitchen interface looking at the purchase of fresh Atlantic salmon (*Salmo salar*) in three convenience grades: purchase of whole salmon (convenience grade 0), purchase of fillets (convenience grade 1) and purchase of portions (convenience grade 2). The data collection for the case study involved questionnaires and expert interviews with the deputy managers of the supplier manufacturing site and with the procurement manager of the hotel kitchen in the course of 2019. Additionally, the authors visited the supplier manufacturing site and one hotel kitchen. Based on the data collected, a calculation model was built in an Excel-based datasheet. This model was subsequently optimised and refined during several months to allow for calculating all the necessary details on the food waste volumes, environmental impacts and costs associated with each convenience grade. During this process, further needs for specific data were identified. These data gaps were then filled in the first half of 2020 through further email, face-to-face and telephone conversations with the fish supplier and the hotel kitchen. All inventory data is listed in the supplementary materials, as described in the next few paragraphs.

A description of the salmon processing chain, from the salmon farm up until the arrival at the supplier and subsequently at the hotel kitchen, is given in Table S1 in the Supplementary Information (SI). For the purpose of this study, it is assumed that both the supplier and the kitchen achieve the same filleting and portioning yield. Filleting of whole salmon results in two fillets and in filleting cut-offs, with a mass-based filleting yield of 62% (Table S2). As a next step, the fish fillets are cut into portions of a standard size. In the case study, one portion of salmon weighs 80g, which is a common serving size for fish at buffets. As such, one fillet yields 13portions. On a mass basis, the salmon portions take up 52% of the initial fish weight, whereby filleting and portioning cut-offs respectively account for 38% and 10%. For each salmon portion weighing 80g, about 58g filleting cut-offs and 16g portioning cut-offs are generated.

Depending on whether an entire fish, fillets or portions are bought, filleting or portioning of salmon takes place at the food service kitchen or at the supplier. This affects what happens with the filleting or portioning cut-offs (Table 2; Table S3). All food that ends up in the bin at the supplier or at the kitchen is collected by a specialised waste company and subsequently used for electricity generation (through anaerobic digestion, AD). The Sankey diagram (Figure 1) illustrates the food and food waste flows for filleting and portioning at the supplier, thus reflecting the situation in which the kitchen procures portions. Sankey diagrams for the situation in which whole salmon or fillets are purchased, can be found in the SI (Figures S1–S3).

**Table 2.** Destination of filleting and portioning cut-offs for each scenario under study (Source: hotel and supplier). All food waste is used for energy production through anaerobic digestion (AD).

| Scenario | Step | Location | Destination of Cut-Offs |
|---|---|---|---|
| Purchase of whole fish (Convenience grade 0) | Filleting | Hotel kitchen | 100% bin (AD) |
| | Portioning | Hotel kitchen | 5% bin (AD); 95% used internally for fish pans or staff meals in the hotel |
| Purchase of fillets (Convenience grade 1) | Filleting | Supplier | 1% bin (AD); 62% valorised as animal feed; 37% used for human consumption |
| | Portioning | Hotel kitchen | 5% bin (AD); 95% used internally for fish pans or staff meals in the hotel |
| Purchase of portions (Convenience grade 2) | Filleting | Supplier | 1% bin (AD); 62% valorised as animal feed; 37% used for human consumption |
| | Portioning | Supplier | 100% used internally for fish pans, terrines, minced fish |

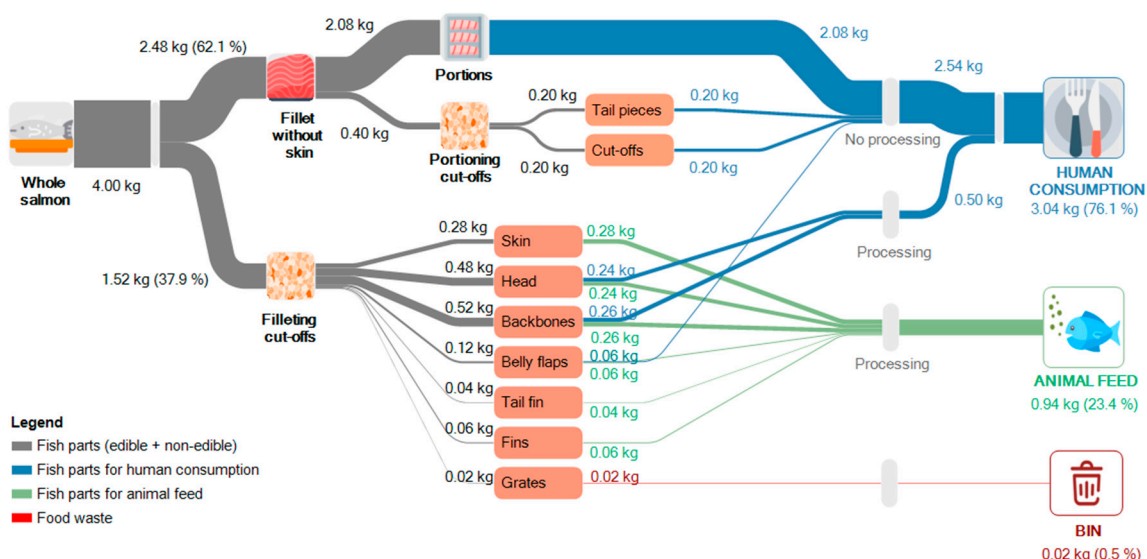

**Figure 1.** Sankey diagram for the purchase of portions: filleting and portioning by the supplier.

## 2.2. Definitions and Food Waste Measures under Study

This paper follows the definition put forward by the EU-funded FUSIONS project [39]. Food waste is hereby understood as "any food, and inedible parts of food, removed from the food supply chain to be recovered or disposed (including composted, crops ploughed in/not harvested, anaerobic digestion, bio-energy production, co-generation, incineration, disposal to sewer, landfill or discarded to sea)".

The present paper assesses how the convenience grade of salmon affects food waste along the salmon chain. To do so, the following three scenarios are investigated (Table 3): purchase of whole salmon (CONV_0), purchase of fillets (CONV_1) and purchase of portions (CONV_2). For the purpose of this study, the situation in which whole salmon is procured (CONV_0) serves as a reference scenario. The two food waste measures under study then refer to (a) procuring fillets (CONV_1) instead of whole salmon, and (b) procuring portions (CONV_2) instead of whole salmon.

Currently, the hotel kitchen purchases a combination of whole salmon, fillets and portions. As such, an additional scenario, Business as Usual (BAU), is defined based on the 2018 purchasing volumes of the kitchen under study. Following the focus of this research, related results are given in

the Supplementary Information only (Section S3), whereas the article itself focusses on using CONV_0 as reference scenario.

**Table 3.** Scenarios under study.

| Scenario | Description | Unit |
|---|---|---|
| CONV_0 | Purchase of whole fish (Convenience grade 0) | Expressed per portion or per year |
| CONV_1 | Purchase of fillets (Convenience grade 1) | |
| CONV_2 | Purchase of portions (Convenience grade 2) | |
| BAU | Business As Usual. Represents the situation of 2018, whereby the kitchen under study purchased a combination of whole salmon, fillets and portions. | Expressed per year |

The food waste measures target the share of preparation waste related to filleting and portioning salmon, namely the filleting and/or portioning cut-offs. Filleting and portioning cut-offs removed from the supply chain and valorised as animal feed, for example, are not considered as food waste, but categorised as by-products [39–41]. When talking about food waste in this study, the authors thus refer to the particular stream of salmon parts or by-products that is disposed of in the bin (and sent to an anaerobic digestion facility) at the supplier or at the hotel kitchen. Also included are salmon storage losses at the hotel and at the supplier, which are also disposed of in the bin. Any other food waste stream related to the fish farming stage, the cooking of salmon or plate leftovers are out of the scope of this study, because they have no influence on these waste reduction scenarios.

*2.3. Sustainability Assessment Framework for Evaluating Measures*

The evaluation follows the methodology outlined in [11,12]. A food waste prevention measure is hereby evaluated based on its effectiveness (food waste reduction potential) and its sustainability across the environmental, the economic and the social dimension (Figure 2). To evaluate the effectiveness of a food waste measure, food waste reductions along the entire chain are assessed (see further in Section 2.4.2). The environmental and economic assessment take into account embodied impacts or costs of food that are no longer wasted and the associated avoided disposal impacts or costs, complemented with the impacts and costs specifically related to the implementation of the measure (see Section 2.4.3). Following the focus of the food waste measures, all aspects associated with filleting or portioning are considered as "implementation impacts or costs". For the social pillar of the sustainability assessment, implementation impacts relate to, for example, meals donated or jobs created.

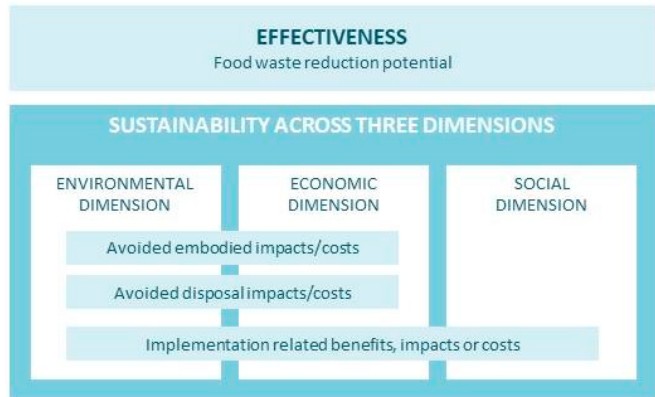

**Figure 2.** Evaluation framework with the different elements taken up in the assessment, adapted from Goossens et al. [11].

*2.4. Application of the Sustainability Assessment Framework to the Case Study*

2.4.1. Functional Unit

Results are reported using two functional units (FU). Firstly, using a portion of 80 g as FU, results are expressed per portion. Secondly, the number of portions served during one year is used as a FU, allowing us to report on an annual basis: in 2018, the hotel chain served around 125,000 portions (Tables S4 and S5). Rescaling the results per portion to an annual basis will make the effectiveness and sustainability of the food waste measures clearer.

2.4.2. Effectiveness

Based on the destination of the filleting and portioning cut-offs (Table 2), the filleting and portioning cut-offs that end up in the bin are calculated for each scenario. This value is complemented with the applicable storage losses at the supplier and in the kitchen to get a full insight into the food waste associated with each scenario. From there, food waste reductions brought about by the measure can be calculated.

The waste volumes generated in each of the three main scenarios (CONV_0, CONV_1 and CONV_2) is then set in comparison to the total volumes of whole salmon, fillets or portions purchased annually (Table S5). As the food waste reductions to be achieved by these two measures (switch from CONV_0 to CONV_1 resp. CONV_2) refer to diverting salmon by-products to animal feed production, for example, rather than throwing them in the bin, the cascade index put forward by the Flemish Department of Agriculture is further used to score the level of valorisation of the measures [42,43]. This index weighs the edible and inedible food parts removed from the food supply chain according to their position on the food waste cascade: flows destined for animal feed production receive the maximum weighting coefficient, "10", whereas those destined for energy production through anaerobic digestion receive an "8" (Table S6).

2.4.3. Sustainability across Three Dimensions

(a) Environmental dimension: carbon footprint

The environmental assessment considers all impacts generated throughout the chain, up until the arrival, storage and eventual filleting or portioning in the kitchen (Table 4; Figure 3). All steps thereafter—such as the preparation and serving of food, as well as plate leftovers—are out of the scope of this paper. Any food waste arising during the fish farming stage and its subsequent transport to the fish supplier is excluded as well. All impact calculations take into account storage losses at the supplier and hotel levels: as an example, for one salmon to be sold to the kitchen, more than one salmon needs to be produced and transported to the supplier to account for supplier storage losses.

**Table 4.** Steps included in the environmental assessment. Categorisation into three impact elements.

| Impact Elements | Step | Description |
|---|---|---|
| Embodied impacts | a | Aquaculture (fish farming in Norway). |
| | b | Transport to the supplier manufacturing site (excl. tertiary packaging). |
| | c | Packaging materials: secondary packaging (reusable plastic crate, ice cubes, plastic cover sheet); no individual primary packaging applicable. |
| | d | Electricity use for storage at the supplier. |
| | e | Refrigerated transport from supplier manufacturing site to its distribution centres, and from there to the hotel kitchens. |
| | f | Electricity use for storage in the hotel kitchen. |
| | g | Disposal of packaging: plastic sheet disposal at the hotel; disposal of reusable plastic crates at the supplier (taking into account reuse rate). |
| Food waste disposal impacts | h | Disposal of storage losses at supplier and hotel kitchen. |
| | i | Disposal of filleting and portioning cut-offs at supplier and hotel kitchen. |
| Implementation impacts | j | Use of filleting/portioning machine at the supplier (electricity use, excl. capital good). |
| | k | Use of water at supplier or at the hotel during filleting/portioning and for cleaning afterwards. |

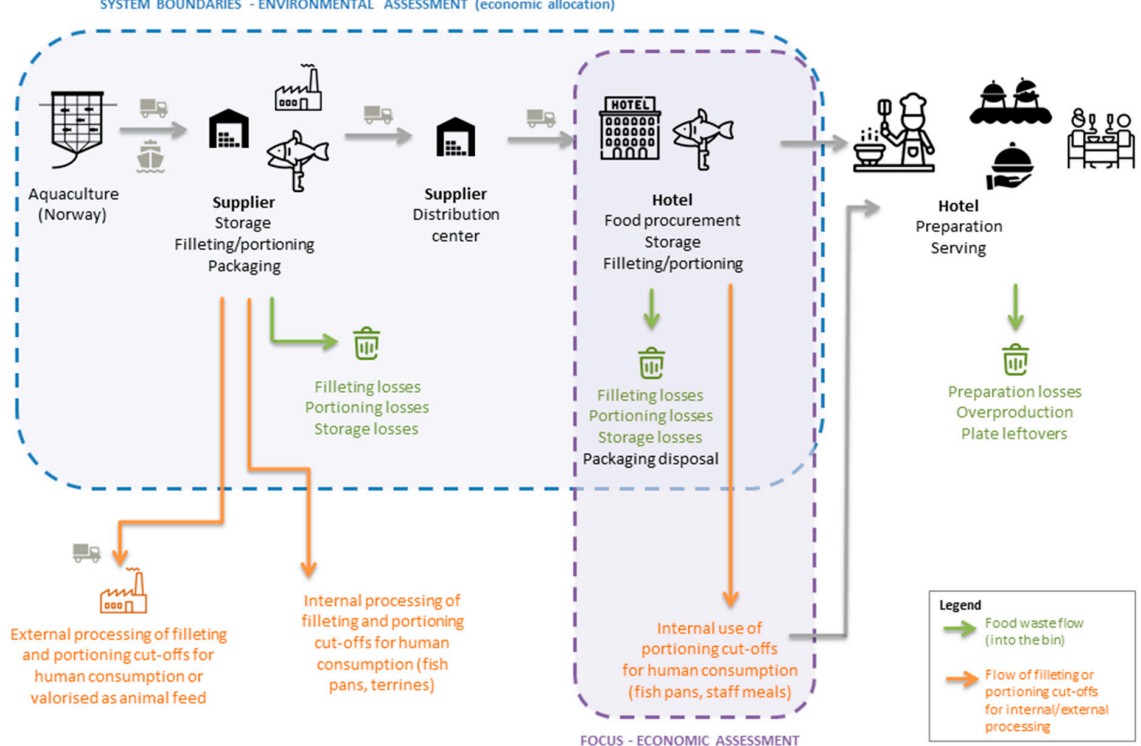

**Figure 3.** System boundaries and focus of the environmental and economic assessment.

The life cycle analysis (LCA) performed as part of the environmental assessment is based on the impact assessment framework of the International Reference Life Cycle Data System (ILCD). Focus is given to the calculation of climate change impacts, expressed as $CO_2$ equivalents, resulting in a carbon footprint calculation of the system under study. To perform the life cycle assessment, additional inventory data—complementing Section 2.1—was obtained from the supplier and the hotel and completed with literature data (Tables S7–S9). Economic allocation is applied to allocate the environmental burdens between the fish portions and the cut-offs (SI, page 11). A contribution analysis is performed to get more insight in those steps that contribute most to the environmental impact along the salmon chain.

(b)   Economic Dimension

The economic cost calculations focus on costs borne by the hotel (Table 5; Figure 3). All costs occurring in any of the previous steps of the food chain (such as staff costs, use of electricity and water, or equipment investments and maintenance at the supplier) are assumed to be reflected by the commodity price; these underlying costs therefore do not appear in the table. The assessment takes into account the costs associated with the internal use of portioning cut-offs in the hotel kitchen, the reason being that the portioning of fillets by the hotel generates portioning cut-offs (so called "bits and pieces") which are perfectly edible and of good quality. About 95% of these cut-offs is used internally for fish pans and staff meals (Table 2). In case the hotel outsources the portioning process, these portioning cut-offs stay at the supplier. In that case, the hotel would need to buy such bits and pieces to be able to make an equal amount of fish pans. In order to take this into account, the purchase of such portioning cut-offs is taken into consideration each time pre-portioned salmon is procured.

For the economic assessment as well, storage losses were taken into account. Additional inventory data needed for the economic assessment is listed in the SI (Tables S7–S9). A contribution analysis was performed to analyse the costs borne by the hotel.

**Table 5.** Costs included in the economic assessment. Categorisation into three cost elements.

| Cost Elements | Step | Description |
|---|---|---|
| Embodied costs | a | Purchase of food (commodity purchasing price) by the hotel. |
| | b | Storage in fridge at the hotel. |
| | c | Disposal of packaging materials at the hotel. |
| Food waste disposal costs | d | Disposal of storage losses at hotel kitchen. |
| | e | Disposal of filleting and portioning cut-offs at hotel kitchen. |
| Implementation costs | f | Labour costs for manual filleting/portioning at the hotel kitchen. |
| | g | Use of water at hotel for cleaning the filleting/portioning workspace. |
| | h | Net costs for the hotel associated with internal use of portioning cut-offs for fish pans as compared to purchasing them from the supplier in case portioning is outsourced to the supplier |

*Product price sensitivity*—Following the contribution analysis of the various cost elements to the total costs borne by the hotel, a sensitivity analysis is performed. The main contributing factor was found to be the purchasing price of whole salmon, fillets and portions. As such, a sensitivity analysis is made on how a 10% change in fillet and portion price affects the net costs associated with the new purchasing scenarios CONV_1 and CONV_2. These costs are then compared to the CONV_0 scenario (with unchanged purchasing prices for whole salmon), in order to find the net cost balance associated with the food waste measure. As a switch to procuring portioned salmon (CONV_2) was found to lead to additional costs for the hotel rather than savings, this paper further looks into the extent to which the portion price would need to decrease for the measure to be profitable for the hotel.

*Influence of staff skills (reflected in labour costs)*—Since labour costs are considered crucial for business decisions, we look at the influence of having staff with greater filleting skills, assumed to be reflected by higher labour costs. The current labour costs are a representative value provided by the hotel itself. These are based on the average staff costs for regular kitchen staff that does not necessarily have highly specialised filleting skills. According to the hotel, staff with specialised filleting skills is hard to find and costs more. However, specialised staff would allow for time savings in filleting and portioning the fish. The authors therefore assess the effect of deploying staff with more specialised filleting and portioning skills on the annual hotel-borne costs. To do so, a 50% increase of labour costs is applied, while a decrease in time spent for filleting (−15%) and portioning (−10%) is assumed. These changes are applied to all scenarios, including the CONV_0 scenario.

(c)   Social dimension

For the social assessment, this paper looks into how a switch towards fish with a higher convenience grade affects meal donation, jobs (or job creation) and the working environment.

## 3. Results

### 3.1. Effectiveness

*Food waste volumes per portion*—In all three scenarios, there are about 74 g of filleting and portioning cut-offs associated with one portion. In the case where the hotel purchases whole salmon (CONV_0), all filleting and portioning takes place at the hotel. In this case, 80% of the cut-offs ends up as food waste in the bin (Table 6). In the next scenario, where the kitchen buys fillets from its supplier (CONV_1), almost all cut-offs are used for human consumption or valorised as animal feed resulting in only 2% of all filleting and portioning cut-offs being thrown in the bin by the hotel or its supplier. This percentage further goes down to only 1% if the hotel directly buys portions from its supplier (CONV_2). Including the storage losses, the total amount of food waste along the salmon chain is at its highest in CONV_0, amounting to 67 g of food waste for each portion of 80 g. In CONV_1, the amount of food waste per portion is already a lot lower at about 7 g, and it further decreases to 4 g for CONV_2.

**Table 6.** Food waste volumes, environmental impacts and costs associated with each salmon purchasing scenario, expressed per portion.

| | | | CONV_0 | CONV_1 | CONV_2 |
|---|---|---|---|---|---|
| | | | Whole Salmon | Fillet | Portion |
| Food Waste Volumes | Filleting and portioning cut-offs thrown in bin | g/portion | 59.08 | 1.55 | 0.77 |
| | Storage losses, thrown in bin | g/portion | 7.96 | 5.53 | 3.19 |
| | Total food waste along the chain | g/portion | 67.05 | 7.08 | 3.96 |
| | Destination of the filleting and portioning cut-offs: kept within the food chain (human consumption), | % | 20% | 49% | 50% |
| | removed from the food chain and valorised as animal feed | % | 0% | 49% | 49% |
| | removed from the food chain and thrown in the bin | % | 80% | 2% | 1% |
| Environmental Assessment | Embodied impacts | g $CO_2$ eq/portion | 444.30 | 372.01 | 373.03 |
| | Disposal impacts | g $CO_2$ eq/portion | $-4.91 \times 10^{-03}$ | $-5.31 \times 10^{-04}$ | $-2.97 \times 10^{-04}$ |
| | Implementation impacts | g $CO_2$ eq/portion | $3.74 \times 10^{-03}$ | 0.56 | 1.11 |
| | Total climate change impacts | g $CO_2$ eq/portion | 444.29 | 372.57 | 374.13 |
| Economic Assessment | Embodied costs | €/portion | 1.49 | 1.36 | 1.72 |
| | Disposal costs | €/portion | $6.22 \times 10^{-03}$ | $4.52 \times 10^{-04}$ | $1.55 \times 10^{-04}$ |
| | Implementation costs | €/portion | 0.18 | 0.10 | 0.08 |
| | Total costs | €/portion | 1.67 | 1.45 | 1.79 |

*Food waste volumes generated annually*—The total annual food waste along the chain is at its highest in the scenario where whole salmon is purchased (CONV_0), mounting to almost 9 tonnes per year (Table 7). Lower food waste volumes are achieved when buying fillets (CONV_1) or portions (CONV_2), with food waste volumes being reduced to less than 1 tonne per year. Moving from buying whole salmon to buying fillets would thus lead to food waste reductions of over 7 tonnes per year, which is a decrease of 89% (Table 8). Even greater food waste reductions of 94% are possible when switching to buying portions. The majority of the savings hereby relate to reducing the amount of filleting and portioning cut-offs that are binned. Instead, the cut-offs are used for human consumption and valorised as animal feed (Figure 4). For a comparison with the BAU scenario, the authors refer to the Supplementary Information (Section S3).

**Table 7.** Food waste volumes, environmental impacts and costs associated with each salmon purchasing scenario, expressed per year.

| | | | CONV_0 | CONV_1 | CONV_2 | BAU |
|---|---|---|---|---|---|---|
| | | | Whole Salmon | Fillet | Portion | |
| Food Waste Volumes | Total food waste along the chain | kg/year | 8753 | 924 | 506 | 2073 |
| | Amount of food waste set in comparison to what is purchased | g food waste/kg food | 436 | 74 | 49 | 154 |
| | Cascade index | | 8.00 | 9.67 | 9.80 | 9.32 |
| Environmental Assessment | Total climate change impacts | kg $CO_2$ eq/year | 58,003 | 48,639 | 47,847 | 49,999 |
| Economic Assessment | Total costs | €/year | 218,307 | 189,860 | 229,527 | 196,048 |

Comparing these annual food waste amounts with the annual purchasing volumes of whole fish, fillets or portions of salmon (Table S5), the amount of food waste per kg salmon purchased decreased from 436 g in CONV_0 to 74 g in CONV_1 and to 49 g in CONV_2 (Table 7). Through an increased valorisation of the food parts removed from the food supply chain as animal feed, the food waste cascade index rose from 8 in CONV_0 to 9.67 in CONV_1 and further up to 9.80 in CONV_2.

**Table 8.** Effectiveness, net environmental impacts and net cost balance associated with the food waste measures under study, using CONV_0 as a reference scenario. Results are expressed as net values per year and as percentage changes. A comparison with the business as usual (BAU) scenario is given in Table S10.

| | | | CONV_1 | CONV_2 |
|---|---|---|---|---|
| | | | Fillet | Portion |
| Effectiveness | Food waste reduction along the chain | kg/year<br>% | −7829<br>−89% | −8247<br>−94% |
| Environmental Assessment | Net environmental impacts | kg $CO_2$ eq/year<br>% | −9364<br>−16% | −10,156<br>−18% |
| Economic Assessment | Net cost balance | €/year<br>% | −28,448<br>−13% | 11,220<br>+5% |

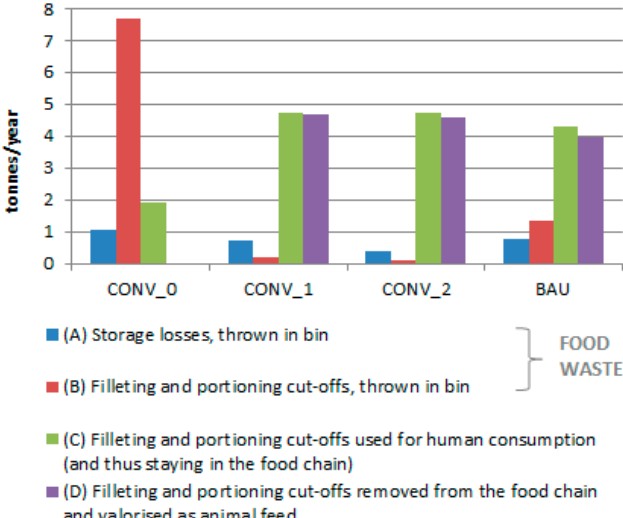

**Figure 4.** Visualisation of the annual amounts of food waste along the salmon chain in each scenario, complemented with the amounts of filleting and portioning cut-offs used for human consumption or valorised as animal feed.

### 3.2. Sustainability across Three Dimensions

3.2.1. Environmental Assessment: Carbon Footprint

*Impacts per portion*—The total climate change impacts range between 0.37 and 0.44 kg $CO_2$ eq per salmon portion of 80 g (Table 6). The highest impacts per portion are found for the situation where the kitchen buys whole salmon (CONV_0). In case the kitchen purchases fillets or portioned salmon, the impacts per portion are almost the same, with those associated with buying portions (CONV_2) being slightly higher than the situation where fillets are procured (CONV_1). It should hereby be noted that these impacts include not only the impacts generated at the level of the hotel kitchen, but also those generated during the previous steps along the salmon chain, such as aquaculture, transport, distribution, storage, packaging and processing at the supplier (as indicated in Figure 3).

The contribution analysis (Table 9) shows that the environmental impacts mainly stem from embodied impacts, with the disposal and implementation impacts playing only a very small role. Based on the contribution of the absolute impact values of each step included in the analysis, the fish farming stage (step a) is responsible for respectively 62% (CONV_0), 73% (CONV_1) and 71% (CONV_2) of the climate change impacts associated with each scenario. In absolute values, the highest fish farming impacts were allocated to one portion in CONV_0 following the economic allocation method. Next in line are the two transport stages related to the transport of whole fish from Norway to the supplier in

Germany (step b) and the transport from the supplier processing site to its distribution centres and from there to the hotels (step e), followed by the packaging stage (step c). Differences in the impacts per portion from distribution, transport and packaging (steps c and e) in CONV_0, CONV_1 and CONV_2 are due to the way the whole salmon, fillets and portions are packaged and distributed (Table S1). In each scenario, the fish, fillets or portions are distributed on a layer of ice, using reusable plastic crates; no individual packaging is applied. In CONV_0, one plastic crate holds one whole salmon, or 26 portions. In CONV_1, one crate holds between 3 and 10 fillets, equal to 91 portions on average. In CONV_2, one crate contains 60 portions. As the packaging and distribution impacts (steps c and e) are calculated per crate, the resulting impacts per portion follow the order CONV_0 > CONV_2 > CONV_1. All other steps (d, f–j) included in the analysis contribute to less than 1% of the environmental impact of the salmon chain.

**Table 9.** Contribution analysis for the environmental assessment. Percentage contribution* of each step along the chain to the total impact of one portion in three situations.

| | | | CONV_0 | CONV_1 | CONV_2 |
|---|---|---|---|---|---|
| | | | Whole Salmon | Fillet | Portion |
| Embodied Impacts | a | Aquaculture | 62.38% | 73.33% | 71.44% |
| | b | Transport to supplier | 14.05% | 16.51% | 16.09% |
| | c | Packaging materials | 4.89% | 1.65% | 2.60% |
| | d | Electricity use: storage at supplier | <1% | <1% | <1% |
| | e | Transport for distribution by supplier: supplier manufacturing site—distribution centre—hotel | 18.60% | 8.33% | 9.53% |
| | f | Electricity use—storage at hotel | <1% | <1% | <1% |
| | g | Disposal of packaging materials | <1% | <1% | <1% |
| Food Waste Disposal Impacts | h | Disposal storage losses, at hotel and supplier | <1% | <1% | <1% |
| | i | Disposal filleting/portioning losses, at hotel and supplier | <1% | <1% | <1% |
| Implementation Impacts | j | Use of filleting/portioning machine at supplier (electricity and water use), incl. cleaning afterwards | 0% | <1% | <1% |
| | k | Manual filleting/portioning at hotel (water use for cleaning) | <1% | <1% | 0% |

*Percentage contribution based on absolute impact values, as suggested by Zampori et al. [44].

*Impacts generated annually*—In case a kitchen procures whole salmon (CONV_0), almost 60 tonnes of $CO_2$ eq per year are emitted along the salmon chain until the arrival and eventual filleting and portioning in the hotel kitchen (Table 7; Figure 5). Switching to procuring filleted salmon (CONV_1) would lead to impact savings of almost 10 tonnes of $CO_2$ eq per year, reflecting a 16% decrease (Table 8). If the hotel would switch to buying portioned salmon (CONV_2), impact savings of 18% would be achieved.

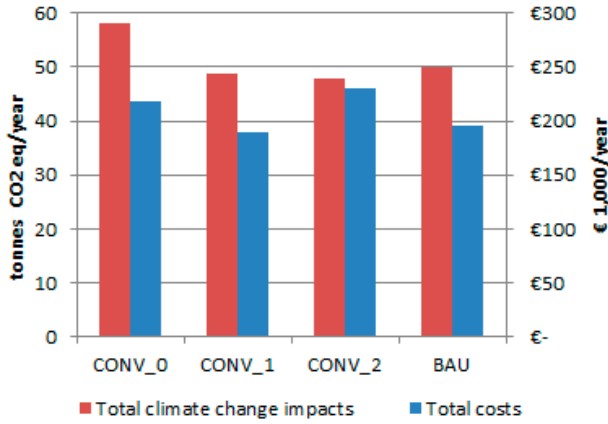

**Figure 5.** Annual climate change impacts and costs associated with each scenario.

The impact savings are mainly due to savings made in the distribution transport and packaging steps (steps e and c in Table 9), following the very high impacts per portion in CONV_0 as compared to CONV_1 and CONV_2, as described above. Other factors contributing greatly to the impact savings are the aquaculture stage and subsequent transport to the supplier (steps a and b). Changes hereby result from differences in total annual storage losses and from the economic allocation method used to allocate the fish farming and transport impacts of whole salmon to one portion in each of the scenarios.

### 3.2.2. Economic Assessment

*Costs per portion*—At a total cost of €1.79 per portion, the purchase of portions (CONV_2) is associated with the highest costs (Table 6). Next in line is the purchase of whole fish (CONV_0), whereas the lowest price per portion is found in the situation where the kitchen purchases fillets from the supplier (CONV_1).

In each situation, the embodied cost elements contribute the most to the total cost per portion (Table 10), with the purchasing price of one portion (based on the price per kilogram of whole salmon, fillets or portions; step a in Table 10) contributing between 89% and 96% to the total costs associated with one portion. The disposal cost elements contribute to less than 1% of the total costs for CONV_0, CONV_1 and CONV_2. However, what matters in the cost breakdown of each situation are the implementation costs. In case the hotel takes up filleting and/or portioning (CONV_0 or CONV_1), the staff labour costs (step f) are of importance. In the case of buying portioned fish (CONV_2), the additional costs for purchasing tail pieces and bits and pieces for fish pans (step h) comes into the picture. In all situations, portioning leads to 15.5 g portioning cut-off per portion. In CONV_0 and CONV_1, 95% of these cut-offs (or 14.8 g) are used internally at the hotel for fish pans and staff meals (the remainder 5% is binned). In CONV_2, portioning cut-offs are generated at the supplier site, not at the hotel. As such, in order to prepare an equal amount of fish pans and staff meals in CONV_2, the hotel would need to buy 14.8 g at the supplier. If the hotel portions its own fillets (CONV_0 and CONV_1), it has its portioning cut-offs directly available and does not have to buy them from suppliers.

**Table 10.** Contribution analysis for the economic assessment. Percentage contribution of each step to the total hotel-borne costs associated with one portion in three situations.

| | | | CONV_0 | CONV_1 | CONV_2 |
|---|---|---|---|---|---|
| | | | Whole Salmon | Fillet | Portion |
| Embodied Costs | a | Purchase of fish (whole fish, fillets or portions) | 88.87% | 93.35% | 95.79% |
| | b | Storage at hotel | <1% | <1% | <1% |
| | c | Disposal packaging plastics (residual waste) | <1% | <1% | <1% |
| Food Waste Disposal Costs | d | Disposal filleting/portioning losses at hotel | <1% | <1% | 0% |
| | e | Disposal storage losses at hotel | <1% | <1% | <1% |
| Implementation Costs | f | Labour costs for filleting/portioning at hotel | 10.73% | 6.61% | 0% |
| | g | Use of water (for cleaning after filleting/portioning) at hotel | <1% | <1% | 0% |
| | h | Purchase of tail pieces and bits & pieces (for fish pans) from supplier in the absence of own portioning cut-offs | 0% | 0% | 4.19% |

*Annual costs*—Purchasing whole salmon costs the hotel kitchen about €218,000 per year (Table 7; Figure 5), whereas procuring fillets or portions costs around €190,000 or €230,000 per year. On an annual basis, the switch to procuring filleted salmon (CONV_1) thus saves the hotel around €28,500 per year, equal to net cost savings of 13%. A switch to purchasing portioned salmon (CONV_2), however, would cost the hotel over €11,000 more, resulting in a 5% cost increase as compared to CONV_0.

The cost savings for switching to CONV_1 result from savings in the purchase of the fillets as compared to purchasing whole salmon, despite the higher prices per kilogram paid for fillets. The reason behind this is that, when purchasing a whole salmon, the kitchen also pays this same purchasing price for the filleting cut-offs that are later on thrown in the hotel kitchen, whereas in the case of fillets, only the fillets are paid for. Other cost savings for switching from CONV_0 to CONV_1 relate

to savings in labour costs, as filleting is outsourced to the supplier. In the case of switching from CONV_0 to CONV_2, the labour cost savings are even higher. Nevertheless, following the high price per kilogram paid for portioned salmon as compared to whole salmon, there is still a significant cost increase associated with this switch. Additionally, when purchasing portioned salmon, new costs arise for purchasing tail pieces and bits and pieces in order to make fish pans. Purchasing these from the supplier contributes to about 4% of the annual hotel-borne costs for CONV_2.

*Product price sensitivity*—In the initial calculations, a switch to procuring fillets resulted in 13% cost savings for the hotel kitchen. If the fillet price would go down by 10%, the change to buying fillets would become even more interesting, leading to cost savings of 21% (Table 11). A 10% increase in the commodity purchasing price of fillets would result in higher net costs for the hotel. Nevertheless, a switch to buying fillets would still lead to cost savings (−5%) as compared to buying whole salmon. When it comes to portions, the initial calculations showed that a switch to buying portions led to additional costs (+5%) for the kitchen. Higher portion prices would only make this effect stronger (Table 11). A 10% decrease in the portion purchasing price would lead to cost savings of 5%, making the switch to procuring portions profitable. It was further found that the net cost balance would drop below zero at a portion price decrease of 5.12%. As such, it can be said that the intervention of moving towards buying portioned salmon instead of whole salmon would be profitable to the hotel kitchen as soon as the portion prices are lowered by about 5%.

**Table 11.** Product price sensitivity (changes in fillet and portion purchasing prices) and influence of staff filleting skills (reflected by 50% higher labour costs). Annual costs and net cost balance for moving from purchasing whole salmon (CONV_0) to fillets (CONV_1) or portions (CONV_2).

| | CONV_0 | CONV_1 | | CONV_2 | |
|---|---|---|---|---|---|
| | Annual Costs (€/year) | Annual Costs (€/year) | Net Cost Balance (%) | Annual Costs (€/year) | Net Cost Balance (%) |
| Initial calculations | 218,307 | 189,860 | −13% | 229,527 | +5% |
| Higher fillet/portion price (+10%) | 218,307 | 207,583 | −5% | 251,514 | +15% |
| Lower fillet/portion price (−10%) | 218,307 | 172,136 | −21% | 207,541 | −5% |
| Lower portion price (−5.12%) | 218,307 | n.a. | n.a. | 218,273 | 0% |
| Improved filleting skills (+50% labour costs) | 226,258 | 194,630 | −14% | 229,527 | +1% |

*Influence of staff skills (reflected in labour costs)*—In the situation where a hotel deploys staff with more specialised filleting skills (+50% labour costs and less time spent for filleting and portioning fish), the annual hotel-borne costs for CONV_0 and CONV_1 increase to about €226,000 and €195,000, respectively (Table 11). As a result, the net cost balance for switching to buying fillets increases, making the measure even more profitable (cost savings of 14% for moving from CONV_0 to CONV_1) than in the initial calculations. When it comes to the purchasing scenario of buying portions (CONV_2), there is no filleting or portioning taking place at the hotel kitchen. As such, the annual costs associated with CONV_2 do not depend on kitchen labour costs and thus remain unchanged. The net cost balance for switching to procuring portions therefore decreases to about €3000 per year, which makes the measure a lot more profitable to the hotel kitchen than in the initial calculations (with less specialised staff). How all of this would affect the job situation in the kitchen is shortly touched upon in the discussion section.

### 3.2.3. Social Assessment

Meal donation is not applicable in the present case study. Even though 8 tonnes of food waste could be avoided on an annual basis in both CONV_1 and CONV_2 (as compared to CONV_0), this mainly refers to better use of filleting and portioning cut-offs within the processing industry, as outlined in Section 3.1. Of those, only the portioning cut-offs and 37% of the filleting cut-offs (Table 2) are fit for human consumption. These are sold, however, by the supplier to external processors, or reused internally. As such, there is no link to be found with food donation.

When it comes to how the food waste measures affect jobs and the working environment, no concrete information could be obtained. As the food waste measures imply, however, that in the new situation no more filleting/portioning needs to be done by the hotel, this might affect the job situation and the working environment in the hotel. Some thoughts on this are given in the Discussion section, based on the informal conversations the authors had with the hotelier and the fish supplier.

## 4. Discussion

### *4.1. Reflections on the Effectiveness of the Food Waste Measures*

#### 4.1.1. Data Inventory: Fish Processing

The data inventory for the processing of salmon was provided by the supplier. This includes the filleting and portioning yield and the percentage fractions of the various salmon by-products as a percentage to the salmon wet weight (Table S2). It should be noted that these by-product percentage contributions differ from those found in the literature. For example, Figure 2 from Stevens et al. shows a contribution of 12.5% for viscera (guts) and 2% for blood [40], whereas these fractions are missing in our data. However, an improved data collection on which by-products exist would not affect the general findings of this study as long as the filleting yield would not be affected. The effectiveness of a reduction measure mainly focusses on whether by-products are thrown in the bin or not, regardless of what these by-products exactly are. Furthermore, as the value of the by-products and the environmental impacts associated with their treatment were found to affect the calculations only to a limited extent, only small changes in the sustainability of the measure can be expected.

Our study assumes zero waste being generated at the external processors, whereas it could be expected that a small percentage of the filleting and portioning cut-offs sent for processing (for human consumption or valorised as animal feed) gets lost during this process or during the transports.

#### 4.1.2. Meal Components Saved

Many studies focussing on reducing food waste calculate how many meals are saved by introducing a food waste intervention [45,46]. In the present paper, only salmon waste is targeted, and therefore it is not possible to calculate the number of whole meals being saved from the bin. Instead, using salmon as the fish component of a meal, we can calculate how many of such meal components were saved from the bin. The food waste measures under study target filleting and portioning cut-offs from salmon. The total food waste reductions brought about by the measures further include reductions in storage losses. In order to see how the present food waste measure and its associated food waste savings contribute to saving meals, only the portioning cut-offs are taken into account, complemented with those filleting cut-offs currently used for unprocessed human consumption (namely the belly flaps). All other filleting cut-offs, as well as the storage losses, are assumed to be unsuitable for human consumption or require additional processing into soups, for example, which lies beyond our present scope.

The calculations have shown that for each salmon portion of 80 g, 73.85 g filleting and portioning cut-offs are generated. Of this, 15.5 g relates to portioning cut-offs that could be used for fish pans or staff meals. In the current situation, 95% (or 14.8 g) of these cut-offs is used for human consumption in CONV_0 and CONV_1, whereas the full 100% is used for human consumption in CONV_2. When it comes to the belly flaps (available after filleting the fish), about 120 g of belly flaps arise per fish. If filleting takes place at the supplier (CONV_1 and CONV_2), 50% of these belly flaps are used for human consumption (equalling 60 g per fish or 2.31 g per portion), whereas the other 50% is valorised as animal feed. In CONV_0, all belly flaps are thrown in the bin. As such, moving away from using non-convenience fish (meaning whole salmon) to procuring fillets (CONV_1) or portions (CONV_2) would instead increase the share of portioning cut-offs and belly flaps being used for human

consumption. The higher the convenience grade of the fish bought, the higher the share of portioning cut-offs and belly flaps used for human consumption.

On a portion basis, changes seem irrelevant: per portion, about 3 g of portioning cut-offs and belly flaps can be saved from the bin (Table S12). On an annual basis, however, 403 kg of perfectly edible food currently ends up in the bin in CONV_0. Moving towards CONV_1, would decrease the amount of edible food thrown out to 101 kg/year, whereas a switch to CONV_2 would completely prevent all this edible food from landing in the bin. This is mostly because of a better use of portioning cut-offs (100% for human consumption instead of 95%) and of belly flaps, but also because of fewer storage losses along the chain when moving towards a system with centralised filleting and portioning. Assuming a fish serving of 80 g, almost 4800 servings can be saved when procuring fillets (CONV_1) instead of whole salmon (CONV_0). This number goes up to over 5000 servings when purchasing portions (CONV_2). As this number refers to serving fish (rather than serving an entire meal), a total of about 4800 or 5000 meal components was saved by the CONV_1 or CONV_2 food waste measure, respectively.

### 4.2. The Magnitude of the Food Waste Addressed by the Measures and the Potential to Scale up to Other Food Products

It was estimated that the hotels considered in this study dispose, on average, of 9 bins of organic waste (240 L/bin) per week in the BAU scenario. As such, the entire hotel group generates 3235 tonnes of food waste per year for all its 32 hotel sites in Germany (Table 12). The food waste generated by the hotels related to the filleting and portioning of salmon (thus excluding the salmon waste generated at the supplier) amounted to 0.06% of the BAU total food waste volume in the hotel.

**Table 12.** Total amount of food waste arising in the hotel chain under study and the contribution of salmon-related food waste (%).

|  |  | CONV_0 | CONV_1 | CONV_2 | BAU |
|---|---|---|---|---|---|
| Total food waste in the hotel chain under study | tonnes/year | 3242 | 3234 | 3233 | 3235 |
| Share of salmon-related food waste in the hotel chain under study | kg/year | 8550 | 621 | 209 | 1785 |
|  | % | 0.26% | 0.02% | 0.01% | 0.06% |

Assuming that the purchasing scenarios CONV_0, CONV_1 and CONV_2 only affect salmon-related food waste, whereas all other biowaste volumes remain unchanged, the food waste measures examined in this paper would decrease the share of the salmon food waste from 0.26% of all food waste being disposed of on an annual basis in CONV_0 to 0.02% when switching to procuring fillets (CONV_1), or 0.01% when purchasing portioned salmon (CONV_2).

The food waste measures investigated in this paper thus affect only a very small percentage of the total amount of food waste arising in a commercial kitchen. Nevertheless, the concept of filleted or portioned fish applies to other fish species as well. Additionally, it may also apply to other food products available in different convenience grades, such as portioned meat and trimmed and pre-cut vegetables. Using products with a higher convenience grade would contribute to reducing the amounts of preparation waste generated in a kitchen stemming from these food products.

Food waste is generally divided into avoidable and unavoidable food waste. Avoidable food waste can be defined as "still fully fit for human consumption at the time of discarding or would have been edible if they had been eaten in time" whereas unavoidable food waste "usually arises when food is being prepared and is discarded. This mainly encompasses both non-edible constituents (e.g., bones, banana peels or the like) and edible ones (e.g., potato peels)" [47]. According to the literature, between 30 and 50% of food waste in the food service sector is avoidable [7]. The food waste reductions brought about by using products with a higher convenience grade address both avoidable and unavoidable food waste arising in the food service sector. In the case study set out in this paper, the unavoidable food waste includes inedible parts such as the fins and backbones of salmon, whereas avoidable food

waste refers for example to the portioning cut-offs and belly flaps. Whereas most food waste measures in the food service sector tend to focus on reducing avoidable food waste, the present measure thus adds to the potential of food services for reducing their overall food waste volumes by also focussing on their unavoidable food waste.

### 4.3. Reflections on the Sustainability of the Food Waste Measures

#### 4.3.1. Environmental Dimension

Following the lack of studies on the environmental impacts associated with using convenience products in the food service sector, as outlined in the introduction section, it was not possible to compare the results of the present paper with literature findings.

When it comes to scaling up the environmental benefits associated with using convenience products in the food service sector, it could be expected that the environmental benefits found for salmon would also apply to other food products, since a higher convenience grade allows for a better use and valorisation of by-products and trimmings of meat, fish, vegetables and fruits. Nevertheless, due attention is needed when generalising the findings: in the present study there is no individual packaging, and packaging impacts thus do not increase (but instead decrease) when moving to a higher convenience grade. The same goes for electricity use during storage. When it comes to purchasing trimmed and pre-cut fruits and vegetables, however, the situation could be different.

#### 4.3.2. Economic Dimension

The model used for calculating the various scenarios in this paper is based on representative commodity prices for 2018 for whole salmon, fillets and portions put forward by the supplier. It needs to be said that purchasing prices are volatile and tend to vary throughout the year, based on fish availability and demand. The profitability of the proposed interventions for the hotel is highly dependent on these purchasing prices.

Furthermore, there are other aspects that may also affect profitability but were not taken into account in the calculations. For a hotel kitchen to fillet and portion fish themselves, the non-specialised staff would have to be trained accordingly. As kitchen staff frequently rotates, such a training investment would need to be repeated every few months. The costs for training these non-specialised staff were not taken into account in the economic cost calculations above. Non-trained staff performing filleting and portioning of fish would further lead to poorly filleted fish, resulting in lower filleting yields and less precise portions. Potential differences in the filleting yield at the supplier and hotel kitchen were not taken into account in the sustainability calculations. Neither were eventual differences in the aesthetics of the portions or eventual differences in portion shapes/sizes leading to differences in preparation time. It could however be expected that the inclusion of these aspects would lead to higher costs for the hotel when purchasing whole salmon or fillets. However, this would not affect the total hotel-borne costs in the situation where the food waste measure of buying portioned salmon (CONV_2) is applied, resulting in lower net additional costs for the hotel, and maybe even in net cost savings. More investigations are needed to evaluate the net cost balance for CONV_1.

#### 4.3.3. Social Dimension

For the social assessment, no definite information was available on how the measure would affect jobs. However, the authors had informal conversations with the hotelier and supplier, giving some insight into the issue: if the hotel buys filleted or portioned fish, a lower demand for staff with good filleting skills at the hotel kitchen can be expected. On the other hand, there is currently a shortage in staff with such specific skills. This means that the chance that staff with good filleting skills would become redundant in the gastronomy sector and get fired is small, as they are currently not employed in that sector to start with. Instead, they are highly sought after by fish suppliers, for example, who can offer them a more attractive salary. Filleting would then not just be something they do on the side while

working in a hotel kitchen, but a full-time job. In the specific situation where highly specialised—and thus more expensive—staff would currently be employed in the kitchen (as examined in the scenario where labour costs would be 50% higher), it may be more profitable to switch to buying portioned salmon and rely on less specialised—and therefore cheaper—kitchen staff. As long as fish suppliers continue to perform manual filleting alongside machine filleting, the jobs of specialised filleting staff at the supplier may be secured. In this case, eventual job losses in the food service sector could be compensated by job opportunities on the supplier side. The moment the supplier aborts manual filleting, jobs could be lost.

The food waste measures discussed in this paper would lead to time savings in the kitchen as filleting, and possibly even portioning, no longer takes place at the hotel. A switch from buying whole salmon to fillets would save 17 h per year per hotel (or just below 3 min/day). A switch to procuring portions would lead to time savings of 37 h per year per hotel, or 6 min per day. Moving towards a wider range of convenience products in a large-scale kitchen would further save time in the kitchen, simplify tasks and speed up the entire mise-en-place. The optimisation of these processes, by diverting them to a more efficient process at the supplier, could lead to substantial time savings and therefore to job losses in the food service sector which would not necessarily be compensated by jobs in the (highly automated) processing industry. Similar insights regarding time savings, reduction of labour costs and less need for skilled staff are given in the literature [35–37]. A thorough examination of such effects through specific case studies would be an interesting avenue for future research.

Considering the high workload for kitchen staff, a positive effect on their working environment can be expected. Additionally, these freed up timeslots could be used for implementing other food waste measures that may require a few minutes extra per day.

However, optimising the mise-en-place process might negatively affect the attractiveness of working in a kitchen or learning to become a kitchen chef, as also stressed by others [35,37]. This aspect of job attractivity based on tasks to be performed (or not to be performed) also needs to be taken into account in future research when looking at the social effects of a more frequent use of convenience products in a kitchen. Moving towards products with a higher convenience grade (as listed in Table 1) such as ready-made sauces and soups, for example, will have an even greater impact on the attractiveness of working in a kitchen. Moreover, it might affect the quality and taste of the food being served, affecting in turn the extent to which guests enjoy the food and come back. In the long run, this may affect the profitability of the kitchen, despite cost savings made from using these convenience products. It is therefore important to balance these cost savings against potential losses in quality or taste.

*4.4. Contribution of the Food Waste Measures to the Greater Societal Goal of Meeting the SDGs and Moving towards a Circular Economy*

Through its food waste reductions, the food waste measures under study contribute to meeting SDG 12.3 [10] while at the same time reducing the environmental impacts along the food chain. A switch to purchasing fillets rather than whole salmon was found to be profitable to the hotel, whereas additional net costs were found for switching towards portioned salmon. In both cases however, the implementation of these food waste measures could contribute to a better image of the hotel in question following its food waste savings and environmental benefits. The FAO further argues that, even when there is no business case to be made for reducing food waste, there might be an economic case, if we take into account the broader societal benefits associated with, for example, environmental impact reductions [9].

Next to reducing the climate change impacts of salmon consumption in the hotel chain in question, the food waste measures related to using products with a higher convenience grade also lead to a more efficient use of resources. Moreover, they contribute to moving towards a circular economy. As many food processing or preparation steps lead to (in)edible parts of a food product being discarded, the convenience grade of a food product determines where along the chain this organic waste or

by-product arises. Consumer food waste is generally recycled into compost and partly used for energy recovery through anaerobic digestion. Based on the food use hierarchy, the valorisation of this discarded organic material as animal feed, for example, would be better [48]. Whereas it may be hard to achieve valorisation at the level of a private consumer or a food service business, the centralisation of food processing or preparation at the level of the manufacturer or food supplier facilitates using discarded (in)edible parts of a food product as a valuable feedstock for other industrial processes. This is being confirmed by the increased cascade index of the system where the food waste measure is in place. And this is exactly what a circular economy is after.

## 5. Conclusions

### 5.1. Study Results

The present study concentrates on the interface between the food service sector and its food suppliers, focussing on fish products with different convenience grades. Through a case study in a hotel kitchen, two food waste measures based on where in the food chain filleting and portioning of Atlantic salmon takes place were examined. The situation in which whole salmon is procured was used as a reference scenario. The proposed food waste measures relate to the kitchen moving to procuring salmon products with convenience grade 1 (fillets) or grade 2 (portions).

For these two food waste interventions, the effectiveness and sustainability were calculated. First, the results were presented on a per portion basis. Next, using the total amount of portions served by the hotel kitchen in 2018, the results were scaled to an annual basis. As such, the extent to which the food waste measures affect food waste along the food chain, reduce greenhouse gases and save costs became clearer.

To calculate the food waste volumes at the supplier and the hotel, salmon food waste stemming from filleting and portioning of salmon at the hotel and at the supplier facilities was taken into account, complemented with storage losses. In the case where only whole salmon was purchased, the total food waste volumes along the salmon chain mounted to 8753 kg/year. Switching to buying filleted fish would reduce these waste volumes to 924 kg/year. A switch to purchasing portions would further reduce food waste volumes to 506 kg/year. As such, the annual salmon food waste was reduced by 89% or 94% when switching to purchasing fillets or portioned salmon, respectively. The majority of the savings hereby relate to a better use and valorisation of filleting and portioning cut-offs at the level of the supplier, whereas these are binned at the hotel.

When procuring whole salmon, 436 g of salmon food waste is generated per kilogram of salmon purchased. When moving towards buying salmon with a higher convenience grade, this amount goes down to 74 g when purchasing fillets and down to 49 g when purchasing portions. Translating the reductions in edible food waste into the number of meal components saved by the food waste measures, the switch to buying fillets would allow saving almost 4800 fish meal components (one serving weighing 80 g) whereas over 5000 servings would be saved in case portions are procured. As such, besides preventing food from becoming waste through better valorisation, the intervention also leads to edible food waste being saved from the bin and used for human consumption instead.

The environmental assessment covers all the steps of the salmon chain, from the fish farming stage up until the arrival, storage and filleting/portioning at the hotel. All steps thereafter (such as the preparation and serving of fish) are out of scope. Besides leading to large food waste reductions, the measures under study were also found to generate environmental benefits along the salmon chain. On an annual basis, 58 tonnes of $CO_2$ eq are emitted along the salmon chain in the situation where the kitchen purchases whole salmon. A switch to procuring filleted salmon would lead to a 16% reduction of climate change impacts along the salmon chain. Purchasing portions would lead to impact savings of 18%, with less than 48 tonnes of $CO_2$ eq being generated on an annual basis.

The hotel-borne (salmon-related) costs are at about €218,000 per year when buying whole salmon. If the hotel were to buy filleted salmon, costs would decrease by 13%. On the other hand, purchasing

portioned salmon would increase the net costs for the hotel by 5%, making this food waste measure not profitable at first sight. Nevertheless, taking into account society-wide benefits such as reduced workload in the kitchen, time freed up for other tasks (such as implementing other food waste measures, if desired) and reduced environmental impacts, there may be an economic case for switching towards portioned salmon after all. Furthermore, a small price decrease for portioned salmon of about 5% would make the switch profitable, as shown in the sensitivity analysis.

When it comes to the social assessment, no effects could be determined based on the information available. However, the purchase of filleted or portioned fish would lower the need for staff with good filleting skills. Informal conversations with the hotelier and supplier have shown that the impact of the proposed food waste measures on job security at the hotel would be low or even non-existing.

*5.2. Implications and Limitations of the Study*

Even though the food waste measure of switching to salmon with a higher convenience grade affects only a very small percentage of the food waste arising in a commercial kitchen, the results may be similar for other raw products such as other fish species, meat, vegetables and fruit for which convenience products are on the market. As such, switching to products with a higher convenience grade may be a promising measure to fight food waste and increase the sustainability of a food service business.

Nevertheless, these findings should not be overgeneralised, as there may be differences in packaging, for example, leading to an increase of the environmental impact associated with increased convenience grades. Moreover, for the economic assessment, the purchasing price of the product will determine which convenience grade is the most profitable to the kitchen in the long run. When it comes to social effects and quality of the food served, this will greatly depend on the extent to which convenience products are used in the kitchen. It would be interesting to see how similar measures for other food products would affect food waste reductions and sustainability along the food chain.

Additionally, next to case studies examining the effects of using various convenience grades of specific products on food waste and sustainability, future studies could look at the bigger picture. The current trend of using more convenience products also affects how commercial kitchens are organised in terms of logistics, appliances to be bought (or not to be bought), size of storage rooms and fridge and freezer capacity. This, in turn, will affect the costs and environmental impacts associated with a kitchen. Along the same lines, similar aspects on the supplier side are to be taken into account in order to get a full picture along the food chain. Furthermore, as mentioned in the discussion section, the optimisation of the supply chain through the use of convenience products, will greatly affect jobs and working conditions in the food industry and food service business.

**Supplementary Materials:** The following are available online at http://www.mdpi.com/2071-1050/12/16/6613/s1, Tables S1–S12, Figures S1–S3. All Supplementary Information to this article is referred to in the paper as "SI". References used in the SI: [42,49–55]

**Author Contributions:** Conceptualization, Y.G., T.G.S. and M.K.; data curation, Y.G.; formal analysis, Y.G.; funding acquisition, T.G.S.; investigation, Y.G.; methodology, Y.G.; project administration, T.G.S.; resources, T.G.S.; supervision, T.G.S.; validation, Y.G.; visualization, Y.G.; writing—original draft, Y.G.; writing—review & editing, Y.G., T.G.S. and M.K. All authors have read and agreed to the published version of the manuscript.

**Funding:** This paper was written within the context of the German ELoFoS research project on Efficient Lowering of Food waste in the Out-of-home Sector. The project ELoFoS was supported by funds of the Federal Ministry of Food and Agriculture (BMEL) based on a decision of the Parliament of the Federal Republic of Germany via the Federal Office for Agriculture and Food (BLE), under the innovation support programme (funding number 281A103416).

**Acknowledgments:** We would like to thank both Lutz Niemann from Maritim Hotelgesellschaft mbH and Björn Reulecke and Felix Clüver from Deutsche See GmbH for providing us with all the data needed to perform this study and for the insights into the day-to-day business of a hotel kitchen and a fish supplier.

**Conflicts of Interest:** The authors declare no conflict of interest. The funders had no role in the design of the study; in the collection, analyses, or interpretation of data; in the writing of the manuscript, or in the decision to publish the results.

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
