# Peer review of "Evaluation of Food Waste Prevention Measures—The Use of Fish Products in the Food Service Sector"

_sustainability, doi:10.3390/su12166613_

Round 1

Reviewer 1 Report

It is an interesting manuscript. Some comments are next given.

- The main weakness of the manuscript is that it is too complicated. Some details and information which nothing add to the manuscript should be eliminated because confuse the reader. Eliminate also some elements such as Table S1, Table S2, etc.

- I suggest the authors to better define the data collection by properly giving more details.

- I suggest the authors edit the entire text to make it more friendly to the reader.

- It is also suggested to properly shorten the title of the manuscript.

Author Response

Dear reviewer,

Many thanks for your constructive remarks. We adapted our manuscript (and supplementary materials), to answer your concerns. All changes are marked in track changes in the document attached. Note that we received feedback from other reviewers as well, which are also reflected in these changes in the manuscript.

We now provide in Section 2.1 more details on how the data was collected. Which data was collected exactly, is then outlined in the remainder of Section 2.1, with specific links to the Tables in the supplementary materials section.

The data collection process resulted in a detailed data inventory of the salmon food chain, in particular of the life cycle stages at the fish supplier and at the hotel kitchen. To allow for better understanding the LCA impact results and for ensuring reproducibility, a lot of inventory data has been provided in the supplementary materials (such as Table S1 and S2). The paper could also exist without publishing this data. However, we feel it is important to provide the reader with all details to allow for comparability in case similar studies would be performed in the future.

In order to reduce the complexity of the manuscript, the authors tried to further reduce the information presented in the manuscript by moving selected text parts of the methodology section to the supplementary materials:

  • Section 2.4.1 on functional unit is shortened; text moved to SI Section S2.2
  • Section 2.4.3 on environmental assessment methodology: the text now only mentions that economic allocation is applied; details on how this is done are moved to the supplementary section (page 11)

We also re-read and re-edited and shortened the entire text to improve readability and make the text more reader-friendly. Lastly, we shortened the title of the manuscript.  

Best wishes,

The authors

ATTACHED: Revised manuscript with track-changes

Reviewer 2 Report

Dear Authors,
your manuscript entitled “ Evaluation of food waste prevention measures in the out-of-home sector. Can the use of products with a higher convenience grade cut food waste, save costs and reduce environmental impacts?” is interesting and original. I think that your paper could be a useful contribution to improve the knowledge on topic and I appreciate your findings that are interesting and well described.

Overall, I assess your paper as an stimulating work and, therefore, only some suggestions to improve it are provided. Please, see the following indications.
- Introduction (literature review) should be integrated with further references to support findings and enhance the discussion;
- References reported in Discussion paragraph should be described more in-depth in Introduction (literature review).
- Conclusion paragraph should be integrated. Especially, theoretical and practical implications should enforced and limitations of the work evidenced.

Best regards.

Author Response

Dear reviewer,

Many thanks for your constructive remarks. We adapted our manuscript (and supplementary materials), to answer your concerns. All changes are marked in track changes in the document attached. Note that we received feedback from other reviewers as well, which are also reflected in these changes in the manuscript.

To strengthen the literature review in the introduction, several text parts of the discussion section were moved to the introduction section. Now it is more clear how the discussion section builds on this literature review.

We also added some sources (in the introduction) to support our findings and the discussion. In particular, we looked for literature on the use of convenience products in the OoH sector, and its effects on food waste volumes and on sustainability.

Implications of the research have been strengthened in the discussion section, and are now also highlighted at the end of the conclusions section.

Best wishes,

The authors

ATTACHED: Revised manuscript with track-changes

Reviewer 3 Report

Authors in the current study presented an emerging topic of research concerning waste in food sector.

The research is designed appropriately with detailed presentation of methods and clear explanation of the results. Moreover, observations are adequately supported by the discussion comparing results with other scientific data.

Author Response

Dear reviewer,

Many thanks for your positive feedback!

Following the feedback from two other reviewers, we slightly adapted our manuscript by adding a few new references and moving some text parts within the manuscript and/or from the manuscript to the supplementary materials. Additionally, the title was shortened. Content-wise however, no fundamental changes were made. Please find attached, for your information, the adapted manuscript, with all changes marked in track changes.

Best wishes,

The authors

ATTACHED: Revised manuscript with track-changes
